

# Glider data collected during the Algerian Basin Circulation Unmanned Survey from 2014 to 2016

Yuri Cotroneo[a,*], Giuseppe Aulicino[a,b], Simon Ruiz[c], Antonio Sánchez Román[c], Marc Torner Tomàs[d], Ananda Pascual[c], Giannetta Fusco[a,e], Emma Heslop[f], Joaquin Tintoré[c,d] and Giorgio Budillon[a,e]

a    Università degli Studi di Napoli "Parthenope", Centro Direzionale di Napoli, Isola C4, 80143, Napoli, Italy
b    Università Politecnica delle Marche, Via Brecce Bianche, 12, 60131, Ancona, Italy
c    Instituto Mediterráneo de Estudios Avanzados, IMEDEA(CSIC-UIB),
     Carrer de Miquel Marquès, 21, 07190 Esporles, Illes Balears, Spain
d    Balearic Islands Coastal Observing and Forecasting System (SOCIB), Edificio Naorte.
     Bloque A, 2º piso, puerta 3, Parc Bit, 07122 Palma, Spain
e    Consorzio Interuniversitario Nazionale per la Fisica delle Atmosfere e delle Idrosfere, CINFAI,
     Piazza Nicolò Mauruzi, 17, 62029 Tolentino (MC), Italy
f    Intergovernmental Oceanographic Commission of UNESCO, 7, place de Fontenoy 75732 Paris cedex 07, France

*    Corresponding author: Yuri Cotroneo – yuri.cotroneo@uniparthenope.it

**Abstract**

We present data collected in the framework of the Algerian BAsin Circulation Unmanned Survey - ABACUS project. ABACUS main objective is the monitoring of the basin circulation and of the surface and intermediate water masses physical and biological properties in a key region of the Mediterranean Sea circulation. Data have been collected through deep glider cruises in the Western Mediterranean Sea during the autumns of 2014, 2015 and 2016. Glider missions were realized in the Algerian Basin, between the Island of Mallorca and the Algerian Coast. Across the three glider missions, eight repeated transects were obtained which enabled us to investigate the basin scale circulation and the presence of mesoscale structures utilising both the adaptive sampling capabilities of the gliders and the higher resolution of the data.
After collection, all data passed a quality control procedure and were then made available through an unrestricted repository host by the SOCIB Data Centre at https://doi.org/10.25704/b200-3vf5. The actual dataset spans three fall seasons, providing an important contribution to the data collection in the chronically undersampled Algerian Basin.
Temperature and salinity data collected in the first 975 m of the water column allowed us to identify the main water masses and describe their characteristics improving the understanding of the dynamics of the region. On the time scale of the project, data show a large variability at the surface layer and reduced variability at the intermediate and deep layers.
Our measurements have been successfully compared to data previously collected in the area from 1909 to 2011. Results showed similar overall distribution, ranges and variability as the historical data, with no outliers in the surface or deep layers.



## 1    Introduction

The southwestern Mediterranean Sea is an important transit region characterized by the presence of both fresh surface waters coming from the Atlantic (Atlantic Water, hereafter AW) and more saline waters which typically reside in the Mediterranean region (Mediterranean Water, MW). At intermediate levels (400 – 1000 m) there is the Levantine Intermediate Water (LIW), typified by subsurface temperature and salinity maxima, while deeper layers (>1000 m) are occupied by the Western Mediterranean Deep Water (WMDW) (Millot, 1999; Millot et al., 2006). Most of the Western Mediterranean is occupied by the Algerian Basin (AB), a wide and deep basin comprised between the Balearic Islands, the Algerian Coast and the Sardinia Channel, where an intense inflow/outflow regime exists and complex circulation patterns take place (e.g., Pascual et al., 2013; Cotroneo et al., 2016; Aulicino et al., 2018). Typically, AW entering through the Strait of Gibraltar flows eastward at the AB surface, mainly inside the Algerian Current (AC), while more saline MW, formed in the eastern and northern parts of the Mediterranean, flows westward at the intermediate and deep layers (Millot, 1985; Testor et al., 2005). As previously demonstrated by several studies, AW and MW interact at different scales, from basin-scale to mesoscale and sub-mesoscale (Robinson and Golnaraghi, 1994; Fusco et al., 2003; Vidal-Vijande et al., 2011), allowing a high seasonal and interannual variability in the basin. This aspect is favoured by the presence of the AC. After leaving the Alboran Sea (Tintoré et al., 1991), this 30-50 Km wide along-slope current flows eastward along the Algerian coast, carrying the AW eastwards (Testor et al., 2005). Typically, the AC becomes unstable along its path due to complex hydrodynamic processes, and forms several meanders which frequently evolves to isolated cyclonic and anticyclonic mesoscale eddies (e.g., Millot, 1985; Moran et al., 2001; Ruiz et al., 2002; Font et al., 2004; Escudier et al., 2016; Cotroneo et al., 2016; Pessini et al., 2018) promoting an intense mesoscale activity all over the AB. These structures present high levels of kinetic energy (Pascual et al., 2013; Escudier, 2016) and impact the distribution of physical and chemical properties of water masses, especially at surface and intermediate depths (Taupier-Letage et al., 2003; Olita et al., 2011).

In the last two decades, both satellites and numerical simulations data have been largely used to study mesoscale processes, partially balancing the scarcity of in situ observations. This improved our knowledge of the large-scale surface features (Vignudelli et al., 2003; Isern-Fontanet et al., 2016), but a complete understanding of mesoscale and submesoscale processes in the basin is still needed. To this aim, frequent dedicated observations at higher horizontal and vertical resolution, along the water column, are essential (Pascual et al., 2017). Multi-platform monitoring strategies, which enable the integration of data from satellites, Autonomous Underwater Vehicles (AUV) and numerical models, have already demonstrated their capabilities in the assessment of oceanographic processes in different regions of the global ocean, such as the Atlantic Ocean (Shcherbina et al., 2015) and the Mediterranean Sea (Carret et al., 2018; Troupin et al., 2018; Pascual et al., 2017; Aulicino et al., 2016). The value of these is improved when implemented along repeated monitoring lines coincident with the satellite groundtracks (Aulicino et al., 2018). Among other AUVs, measurements at small sampling intervals collected through gliders (<5 Km spatial resolution) have contributed to several ocean studies (Rudnick, 2016, Heslop et al., 2012), from dynamics (e.g., Ruiz et al., 2009; Bosse et al., 2016; Cotroneo et al., 2016; Thomsen et al., 2016) to physical and biogeochemical exchanges (e.g., Ruiz et al., 2012; Bouffard et al., 2010; Cotroneo et al., 2016; Olita et al., 2017) or assessment of altimetry data (Heslop et al., 2017). However, gliders generally have a limited speed (e.g. 25 cm/s in the horizontal) that can give synopticity problems when monitoring mesoscale phenomena (Rudnick and Cole, 2011; Alvarez and Mourre, 2012; Aulicino et al., 2016; Cotroneo et al., 2016) or wide areas (Liblik et al., 2016). Thus, their combination with reliable satellite products, numerical simulations and, when available, other



platforms such as drifters, ARGO floats and ship-borne CTDs or a multiple glider missions represent the good strategy for implementing observatories for marine science research which aim to study basin scale and/or coastal processes (Aulicino et al., 2018) where issues of synchronicity are a concern.

Furthermore, this integrated glider-based approach is also important for validating new satellite products, such as the now fully-operational Sentinel-3 mission (Drinkwater and Rebhan, 2007; Heslop et al., 2017) and the upcoming Surface Water and Ocean Topography (SWOT) wide-swath radar interferometer (Fu and Ferrari, 2008), where multi-platform data are expected to help distinguishing noise from small dynamical structures, as a result of cross calibrations between the different sensors (Bouffard et al., 2010; Pascual et al., 2013).

In this study, we present data collected through three SLOCUM G2 (Figure 1) glider missions carried out in the AB during fall 2014, 2015 and 2016.These cruises were developed along a repeated monitoring line set across the AB and allowed the collection of a huge dataset of physical and biological ocean parameters in the first 975 m of the water column. For each mission, glider main track was chosen to lie under overflying altimeter satellites tracks, in order to optimize the inter-comparison of the data from the two platforms. Additionally, during the 2014 and 2016 glider cruises, the sampling track was changed in order to sample specific mesoscale circulation structures in the area. Even though this study is restricted to a particular area, i.e. the AB, we strongly believe that the scientific community could use these data to enlarge the knowledge of the western Mediterranean Sea and to refine the implementation of glider missions and integrated multi-platform observatories in other regions. Glider characteristics, sensors details, paths and mission strategies are described in Section 2; data format and quality control procedures are presented in Section 3. In section 4, a comparison between ABACUS observations and historical data, as well as sample ABACUS transects, are presented. Finally, Section 5 reports main conclusions.

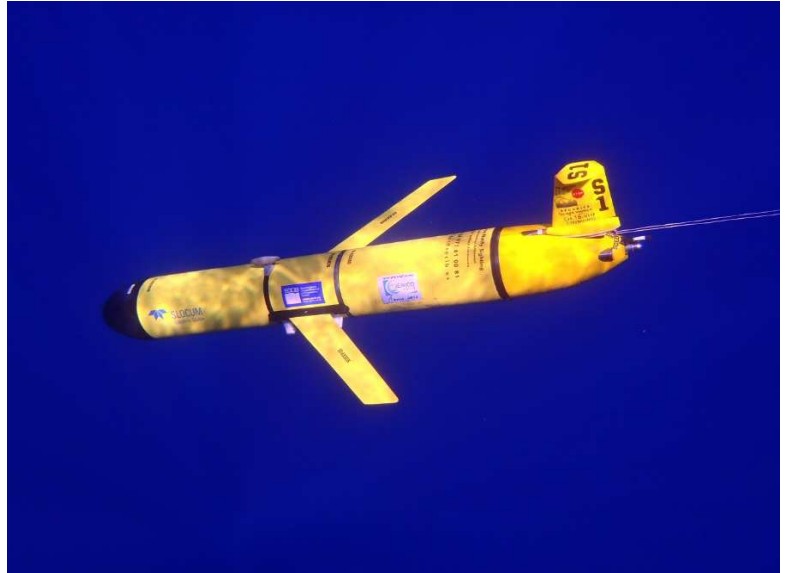

*Figure 1. The* SLOCUM G2 *glider during the ABACUS pre-mission test at sea. Credit M. Torner.*



## 2 Glider field activities and technical details

Gliders are AUV able to provide high resolution hydrographic and bio-chemical measurements.
These vehicles control their buoyancy to allow vertical motion in the water column and make use
of their hydrodynamic shape and small fins to make horizontal motions. Changing the position of
the center of gravity with respect to the center of buoyancy helps controlling both pitch and roll
(Webb et al., 2001). This results in a typical saw-toothed navigation pathway to a maximum depth
of 1000 m.
During the autumns from 2014 to 2016, three deep SLOCUM G2 glider missions were carried out
in the AB (Figure 2) in the framework of the SOCIB (Balearic Islands Coastal Observing and
Forecasting System) Glider Facility Open Access Programme and supported by the Joint European
Research Infrastructure network for Coastal Observatories (JERICO) Trans National Access (TNA).
Funding agreements were developed under the JERCIO TNA third call (grant No 262584) for
ABACUS 1, the SOCIB external access (ABACUS 2) for ABACUS 2 and JERICO-NEXT TNA first call
(grant No. 654410) for ABACUS 3.
Up to 2016, a total of 8 repeated transects were realized between the Island of Mallorca and the
Algerian Coast. In 2014 and 2015, after the realization of the defined transects, the glider track
was changed in order to sample specific mesoscale structures in the study area (Figure 2).

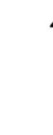

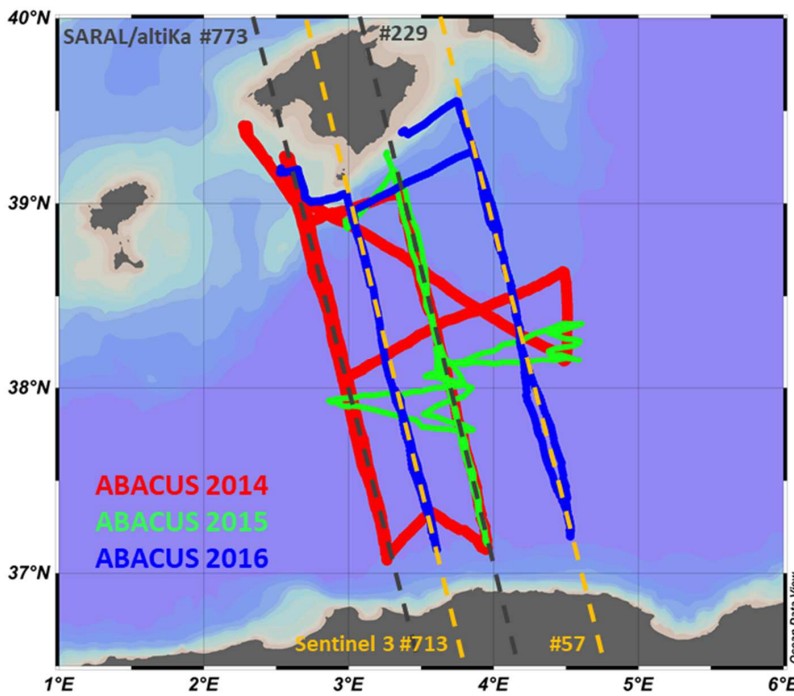

*Figure 2. Glider tracks during the missions of ABACUS 2014 (red dots), 2015 (green dots) and 2016*
*(blue dots). The deviations from the monitoring line were undertaken to sample some mesoscale*
*structures identified through near real time satellite altimetry and SST maps. The groundtracks of*
*the SARAL/altiKa (grey dashed lines) and Sentinel 3 (yellow dashed lines) satellites over the study*
*area are also showed.*



During these cruises ABACUS gliders collected Temperature, Salinity, Turbidity, Oxygen and
Chlorophyll (CHLA) concentration data in the first 975 m of the water column. Each mission had an
average duration of about 40 days and was always performed between September and December.
In situ data collection was supported by remotely sensed data from different platforms over the
Western Mediterranean Sea. In particular, gridded altimetry data provided by the Archiving,
Validation and Interpretation of Satellite Oceanographic data (AVISO), Sea Surface Temperature
(SST) and CHLA concentration information from MODerate resolution Imaging Spectroradiometer
(MODIS) data acquired by NASA were used to provide a large scale description of the dynamics
and surface water masses.ABACUS field activities were performed in collaboration with SOCIB and
the Mediterranean Institute for Advanced Studies (IMEDEA) using a SLOCUM G2 glider for deep
water (1000 m maximum depth) with a vertical speed of 0.18 ± 0.02 m/s, resulting in a horizontal
velocity of about 0.36 m/s. During all missions, the data acquisition design was set to dive with a
descending angle of 26° between 20 m and maximum depth (Figure 3).

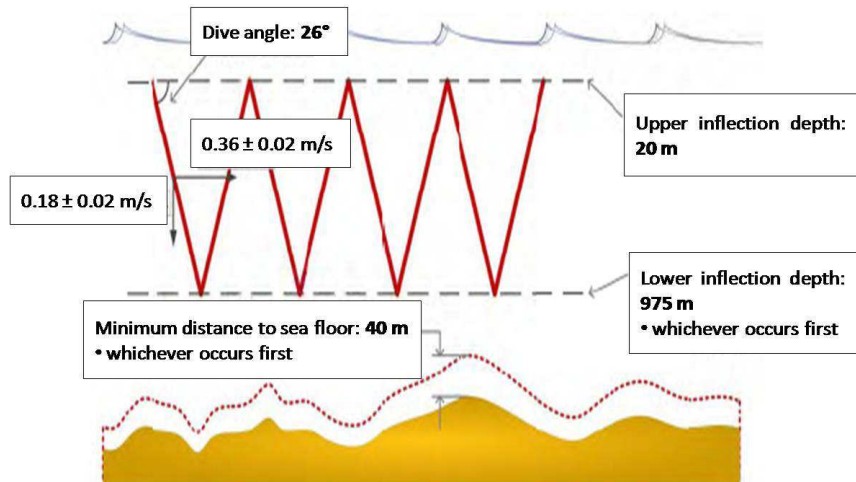

*Figure 3. Glider navigation scheme during the ABACUS missions. The upper inflection depth was*
*changed after the ABACUS 2014 mission in order to allow the glider reaching the surface at every*
*second profile.*
Several physical and optical biochemical sensors were carried aboard the glider for measuring the
ocean temperature, salinity, oxygen, turbidity and CHLA concentration, at different rates
according to depth. In particular, ABACUS gliders were equipped with a glider-customized CTD by
Seabird-Scientific (http://www.seabird.com/glider-payload-ctd) measuring temperature, salinity
(derived from conductivity) and depth, and a two-channel combo Fluorometer-Turbidity sensor by
Wetlabs-Instruments (http://www.seabird.com/eco-puck), both of them embedded inside the
central hull segment of the glider commonly known as the "science-bay". Additionally, an oxygen
optode to measure absolute oxygen concentration and saturation (%) by AADI
(https://www.aanderaa.com/productsdetail.php?Oxygen-Optodes-2) was externally mounted on
the aft, and wet, section of the glider.



The accuracy of the measurements, their vertical resolution and depth range vary according to the
specific instrument and measured variable as reported in Table 1.
Temperature, salinity and oxygen data were sampled to full diving depth (0-975 m depth) while
the acquisition of the other optical parameters ceased at 300 m depth.

| Parameter | Instrument | Sampling rate (Hz) | Vertical resolution (m) | Depth range (m) | Accuracy | Resolution |
|---|---|---|---|---|---|---|
| Temperature (T), Conductivity (C), Depth (D) | Seabird GPCTD Glider payload pumped CTD | 1/2 | 0.4 | -5 to -975 | T ± 0.002 °C<br>C ± 0.0003 S/m<br>D ± 0.1% fsr* | T 0.001 °C<br>C 0.00001 S/m<br>D 0.002% fsr* |
| Oxygen | AADI Optode 5013 | 1/4 | 0.8 | -5 to -975 | <8 µM or 5% | <1 µM |
| Fluorescence (F), Turbidity (Tu) | WetlabsFLNTUslk | 1/8 | 1.6 | -5 to -150 | Sensitivity<br>F 0.015÷0.123 µg/L<br><br>Tu 0.005 ÷0.123 NTU | |
| | | 1/16 | 3.2 | -150 to -300 | | |

*Table 1. Sampling rate and vertical resolution of ABACUS glider data (adapted from Cotroneo et al.,*
*2016; Aulicino et al., 2018). * Full Scale Range*
A specific sampling rate was defined for each instrument (Table 1). Physical parameters
(temperature and salinity) were sampled at 1/2 Hz, resulting in a vertical resolution of 0.4 m along
the water column. Oxygen concentration was acquired at 1/4 Hz (vertical resolution 0.8 m) whilst
turbidity and chlorophyll were sampled at 1/8 Hz until 150 m depth and at 1/16 Hz from that level
until 300 m depth, with a vertical resolution of 1.6 m and 3.2 m respectively.
Calibration processes and regular maintenance were carried out before and after every glider
mission and guarantee the quality of the measurements. While temperature and salinity sensors
were regularly calibrated, unfortunately the optical sensors used during the ABACUS missions
were not calibrated before the cruises. This results into the impossibility to use the derivated-
variable values as absolute values. Nonetheless, gradients along space, depth and time can
successfully be observed and discussed.
Generally, real time data transmission from the glider can be adaptively configured before and
during the mission. For the ABACUS missions, it was set to occur at about every 6 hours, in
correspondence with every second upcast for most of the glider transects. Real time data (having
a file size ranging between 20 and 90 Kb) were transmitted using Iridium satellite link and
contained a subset of all measurements, populated taking one out of every three samples for each
parameter. This strategy permitted the retrieval of a first overview of the data collected, as well as
the eventual transmission of new sampling and navigation directives to the glider. The full
resolution dataset remained stored locally inside the glider until the vehicle was physically brought
back to SOCIB facilities once the mission was completed. Then, data were transferred to SOCIB
Data Center where pre-processing, processing and validation were carried out and NetCDF files
created.
The structure and content of the ABACUS NetCDF file is described in section 3.
The pre-mission activities were carried out at the SOCIB glider facility (Tintoré et al., 2013) and
included all ballasting, pressure tests, compass validation, and adjustment operations needed to
assure the glider capability to reach the surface. This capability is provided by adjusting the overall
weight of the vehicle and by distributing part of that weight so that inclination and roll are neutral



when the glider sets its mechanical actuators (pump, mass shifter, air bladder and fin) in neutral
mode (0 cc, 0 inch, no inflation and 0 rad respectively). That tuning is driven by the hydrographical
characteristics of the target waters to be navigated, i.e. their minimum density, since they
condition the execution of the mentioned operations. Within this scope, the climatological
maximum value of temperature and minimum value of salinity for the studied area and period
have been analysed. These data were used as extreme hydrographic characteristics of the target
waters and allowed us to derive the minimum density (1024.0683 Kg/m$^3$) needed to precisely tune
the glider for the target waters. This tune up is required in order to make sure that the vehicle will
always be able to break into the surface and to raise its tail above the water to have clear and
robust communications.
As regards compass, Merckelbach et al (2008) developed a method to assess the glider compass
error in Slocum gliders. This approach has been followed for gliders used during the later ABACUS
experiments. The methodology should take place at any location away from sources of hard or
soft electromagnetic material (e.g. sport ground), thus it was performed in a forest cleaning to
minimize electromagnetic interference effects. Using a wooden platform, the glider was placed
horizontally and a plastic-made compass stand aligned with the true North was used allowing
precise and repetitive heading gradients of 15 degrees. The measurement error was then
estimated by calculating the difference between the heading measured by the glider and the
heading that was physically imposed using the stand. A maximum error of less than 10 degrees
was observed during the three tests so that compass re-calibration was not considered necessary.
All these operations, as part of the pre-missions tests, contributed in assessing the quality of the
collected data.
The glider tracks were designed to cross the basin from the island of Mallorca (Spain) to about 20
miles off the Algerian coast (Figure 2). The timing of the missions was accurately planned in order
to provide synoptic in situ data with respect to the satellite SARAL/ALtiKa and Sentinel-3A
passages, being also comparable among the different ABACUS missions (Aulicino et al., 2018).
In 2014, during the ABACUS 1 mission, three glider transects were completed along the
neighbouring SARAL/AltiKa groundtracks 229 and 773. The satellite overpassed the glider on 17
September and 26 November 2014 (track 773) and on 12 December 2014 (satellite track 229).
During ABACUS 2 mission, the glider was overflown by the SARAL/AltiKa satellite on 23 October
2015, along the groundtrack 229 with the glider halfway along the transect. In fact, analysis of the
data collected during the 2014 mission, highlighted the need of a higher degree of synopticity
between the glider and satellite data. A specific deployment plan was then designed for the
following ABACUS missions in order to have the glider approximately in the middle of the transect
during the satellite passages. The improvement obtained in synopticity are extensively discussed
in (Cotroneo et al., 2016; Aulicino et al., 2018).
During October 2014, a deviation from the planned sampling track was realized in order to
investigate the presence and sample a mesoscale eddy (see red butterfly shape track in Figure 2).
The eddy presence was first identified on the basis of near real time satellite altimetry and SST,
then the glider track was modified during one of the glider surface communication periods. The
adaptive sampling capabilities of the glider were then successfully used to perform two full depth
transects across the mesoscale eddy (Cotroneo et al., 2016). After that, the scheduled ABACUS
sampling track was restarted. The same strategy was applied in 2015 when the glider deviated
twice from the planned track to monitor the edge of a possible mesoscale eddy (saw-tooth green
track in figure 2).



Finally, during the ABACUS 3 mission, two new glider transects were performed along the Sentinel-
3A groundtracks 57 and 713, the latter being located between the SARAL/AltiKa groundtracks 229
and 773 (see Figure 2). During this mission the glider was overflown on 12 November 2016 (track
57) and 5 December 2016 (track 713).
Resolution of sampling was defined according to the energetic constraints of the platform and to
the scientific aims of the missions, which required high resolution in both horizontal and vertical
directions to monitor large scale, as well as, mesoscale processes. During the first ABACUS mission,
the glider was programmed to sample only during downcasts with a final along-track resolution of
almost 4 km once the oblique profiles are projected into the vertical. As a result of the first data
analysis, the sampling plan for the following missions was modified. During ABACUS 2 (2015) and
ABACUS 3 (2016) cruises, both downcasts and upcasts data were collected, thus obtaining an
improved spatial along-track resolution of about 2 km, while diving angle and speed remained
unchanged. Apart from increasing horizontal sampling resolution, this new strategy also has an
effect on data quality, enabling the application of the thermal lag correction developed by Garau
et al. (2011) which requires consecutive up and down profiles of temperature and salinity.

**3    Data quality control**
After each mission, data were transferred from the internal glider memory to the SOCIB Data
Center, where data processing was carried out and production of delayed time NetCDF files at
different elaboration levels (i.e., level 0, level 1, level 2) occurred (Troupin et al., 2016; Cusi et al.,
2013) before web dissemination of the data.
Each NetCDF file comprises the main cruise information, data and a short abstract. Level 0
contains the raw data collected by the glider without any elaboration or correction and organized
in vectors. Level 1 includes data regularized, corrected and/or derived from raw glider data. Each
variable is stored in a single vector containing all the observations acquired during the cruise and
is described and commented individually. Level 2 dataset includes the regularly sampled vertical
profiles generated from depth binning of already processed (level 1) glider data. In this level, each
variable is stored in matrix. Depth increases with the number of rows, with each row
corresponding to about 1 m depth, while time and consequently the number of profiles changes
along the columns.
Delayed mode data processing for level 1 and 2 included thermal lag correction, filtering and 1 m
bin vertical averaging (Troupin et al., 2016). Nevertheless, some suspicious data were still present
in the surface layer after this standard quality control protocol. The presence of these doubtful
values at the shallower turning depth of the glider supports the hypothesis of a possible effect of
glider navigation phase change on the sampling capability.
For this reason, during the 2015 and 2016 missions, the glider was programmed to reach the
surface after every profile in order to avoid this issue, improve the data collection in the very
surface layer (depth < 20 m) and provide a more suitable dataset for the comparison with satellite
data.
Nevertheless, an additional quality control procedure was developed and regularly performed at
University of Naples "Parthenope" on all ABACUS data. In particular, this procedure includes an
additional single-point spike control, the interpolation of single missing data along the profiles, the
application of a dedicated median filter, a 5-point running mean along the depth and, finally, an
iterative comparison between adjacent profiles. This tool allowed us to identify and discard bad





data and artefacts that were still present after the standard quality control. This final product is
indicated as level 3 of the ABACUS dataset. An example of the level 3 elaboration steps is given in
Figure 4. This additional procedure was routinely adopted for the quality control of all ABACUS
glider missions for each geophysical parameter.
All dataset elaboration levels from 0 to 2 are available at https://doi.org/10.25704/b200-3vf5,
while level 3 data are freely available upon direct request to the authors. Data samples presented
in this manuscript are obtained from level 3 data.

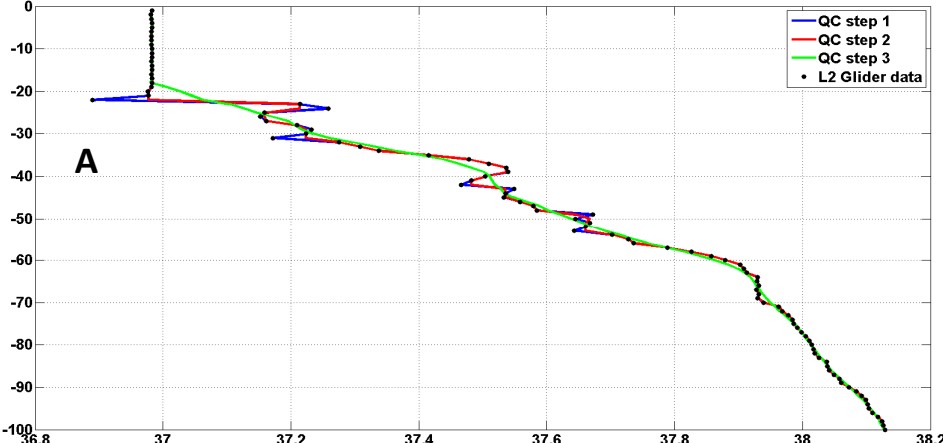


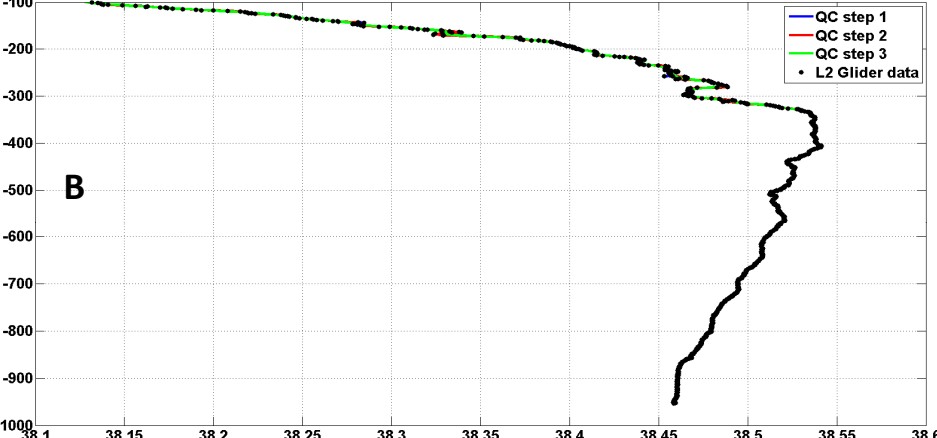


*Figure 4. Effect of the additional quality control used to generate level 3 data. An example is shown*
*for a salinity profile of the ABACUS 2014 mission. The surface layer is shown in panel A (0-100m*
*depth) while the intermediate and deep water layers are shown in panel B (from 100 to 1000 m*
*depth). Black dots represent the original level 2 data after the standard quality control procedures,*
*blue line shows the single missing data interpolation, the median filter effect is shown through the*
*red line, finally the green line represents the effect of the 5-point running mean on the data.*




Figure 5 shows the mean temperature and salinity profile calculated for each transect completed
during the ABACUS cruises from 2014 to 2016. For each mission, a mean profile was calculated
using data of a single Mallorca-Algerian coast (or vice versa) transect. In this figure, the first
available mean profile for each mission is shown in blue, the second in black, the third in red and
the fourth in green. Standard deviation along depth during each transect was also computed for
both temperature and salinity (shaded areas in Figure 5).
The ABACUS profiles are characterized by a high level of variability in the surface layer that
considerably decreases in the intermediate and deeper layers, in agreement with existing
literature in the area (Manca et al., 2004; Fusco et al., 2008). Mean standard deviation values
range between 0.01 and 0.04 for salinity and between 0.05 °C and 0.15 °C for temperature; these
values are generally larger than the precision of the glider sensors previously described (Table 1).
This supports the assumption that glider measurements are accurate enough to represent the
variability of the basin. Ship-based CTD profiles collected at the same time and location of the
ABACUS casts and corrected to bottle samples could provide material for an interesting
comparison between glider and CTD probe data, but unfortunately, no ship-based CTD data were
collected in correspondence of the ABACUS missions up to 2016.




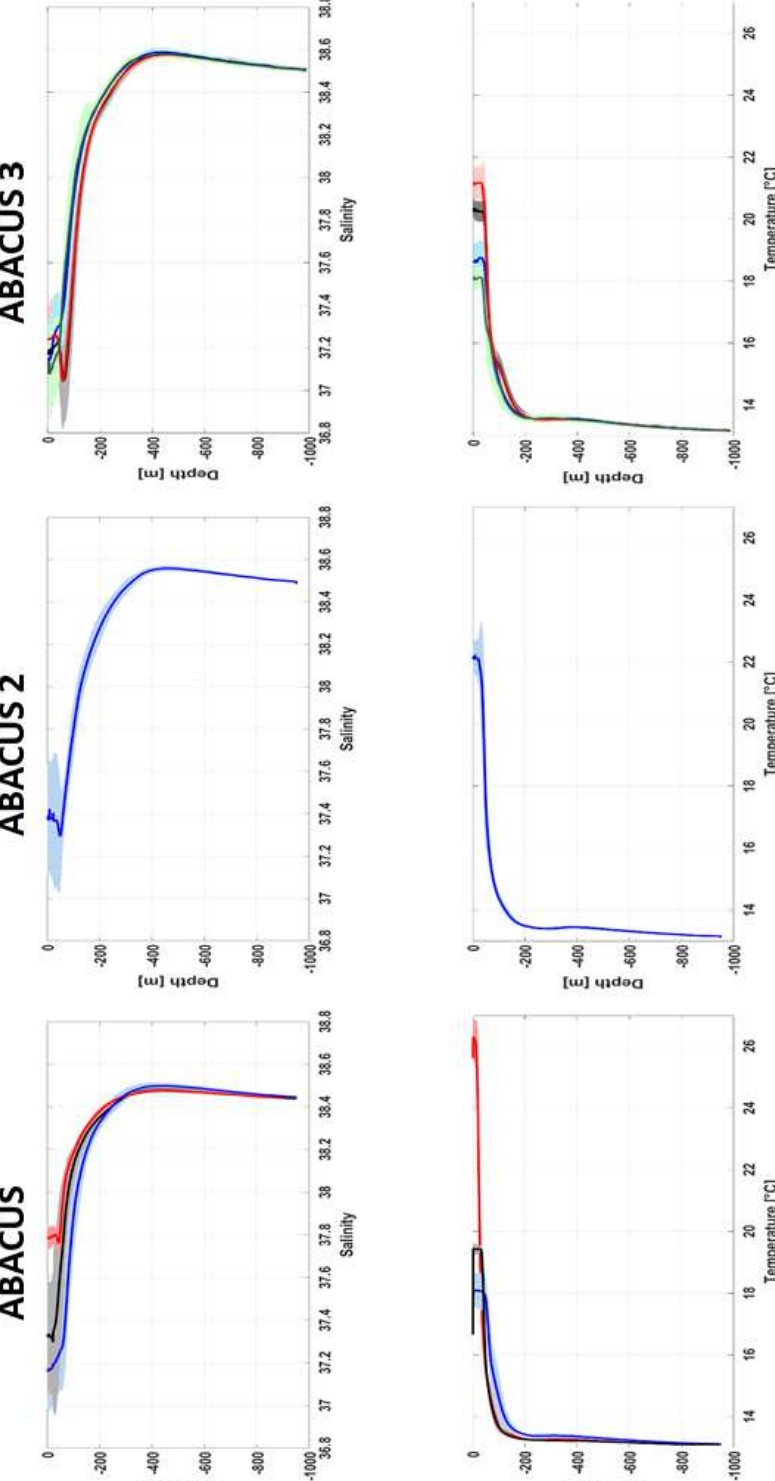

*Figure 5. Mean salinity (upper panels) and temperature (lower panels) profiles (thick lines) for each ABACUS transect. From left to right, ABACUS 1,*
*ABACUS 2 and ABACUS 3 data are represented. For each mission, the first available mean profile is shown in blue, the second in black, the third in*
*red and the fourth in green. Shaded area represents the standard deviation calculated at each depth for each mean profile.*



**4 Data comparison and transect samples**


In order to test the reliability of ABACUS dataset, a comparison was performed between glider
data and a set of historical oceanographic measurements collected in the study area. In particular,
a composite dataset including temperature and salinity data along the water column was realized
merging data from the MedarMedatlas II project (MEDAR group, 2002), from the Coriolis CORA-
3.4 Dataset (Cabanes et al., 2013) and from the World Ocean Hydrographic Profiles (WOHP - V1.0
database, in agreement with Viktor Gouretski). The resulting dataset spans from 1909 to 2011
with a regular distribution of the data across the different seasons. After the application of
standard quality control procedures, the calculation of potential temperature (Θ) was carried out
and the Θ/S couples from this dataset and the ABACUS observations were compared.
Figure 6 shows the comparison between ABACUS Θ/S data and all the available observations in the
study area. ABACUS data have the same distribution of the historical data, with no outliers in the
surface or deep layers. These results confirm that glider ABACUS data captured, and correctly
describe, the main thermohaline properties of the AB water masses and their variability.

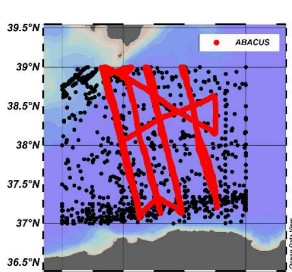
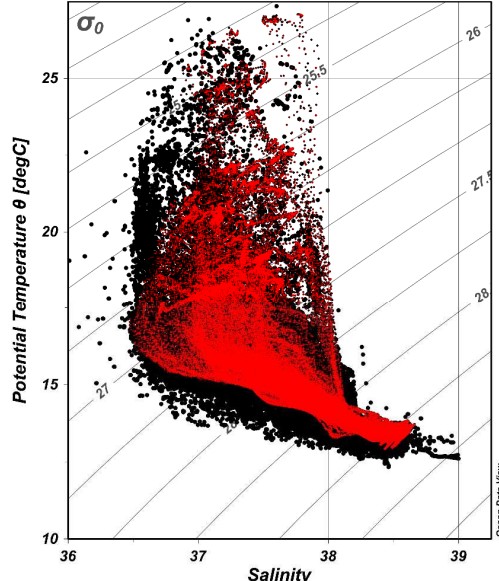

Figure 6. Θ/S diagram comparing historical oceanographic observations from 1909 to 2011 (black
dots) to ABACUS data (red dots). The associated map shows the spatial distribution of the data.

In order to perform an additional and more accurate test on the data reliability, we selected the
data collected during the fall season (September – December) from the merged historical dataset.
This subset was then used for a second comparison with the ABACUS data. Results are showed in
the Θ/S diagram in Figure 7. Again, the ABACUS data successfully represent the hydrographic
variability of the area, even when analysing data from a selected season. Furthermore, the
reduced number of historical data in the fall season and their sparse distribution are a clear
indication of the relative importance of ABACUS dataset for studying the AB.

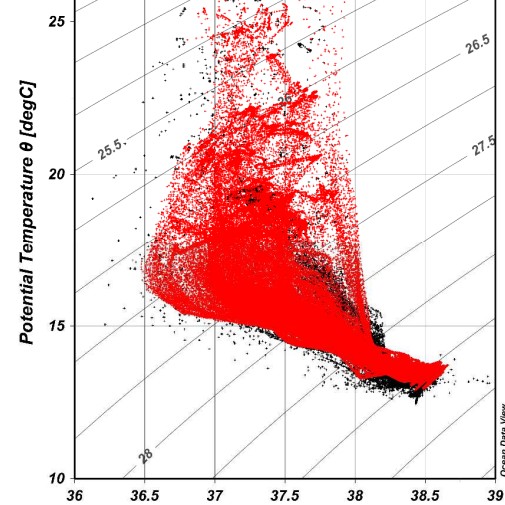

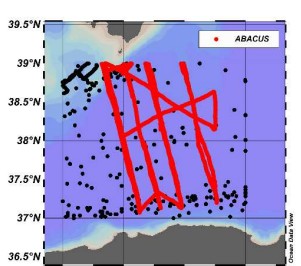

Figure 7. Θ/S diagram comparing historical oceanographic observations (black dots) during the fall season (September – December from 1909 to 2011) to ABACUS data (red dots). The associated map shows the spatial distribution of the data.

Several typical Mediterranean water masses have been identified in the θ-S diagrams derived from the glider mounted CTD during the three ABACUS missions at sea. A summary overview is provided in Table 2. The surface layer (0 to 50 m depth) was occupied by AW whose properties vary greatly according to different stages of mixing, precise geographical position and residence time in the Mediterranean Sea. Potential temperature, for example, ranges between 14.4°C and 27.0°C with colder waters always identified in the southern part of the AB. More mixed and modified waters are present in its northern sector due to the influence of Balearic waters (Cotroneo et al., 2016). As expected, the effect of seasonal cooling can be detected too; in fact, December missions presents lower mean surface water temperatures (below 19°C).

As for salinity, its values in the surface layer range between 36.51 and 38.04, with higher values measured next to the Mallorca Channel, characterized by shallow bottom and easier intrusion of saltier waters (Aulicino et al., 2018).

The intermediate layers were typically occupied by LIW, characterized by a potential temperature generally lower than 13.5 °C, a salinity of about 38.5 and a low oxygen concentration (see an example in Figure 8c). Its presence was mostly identified between 300 and 550 m depth.

The deepest observed layers (between 700 and 1000 m depth) were usually characterized by the presence of WMDW with typical θ values ranging between 12.9°C and 13.2°C and salinity ranging between 38.44 and 38.49. These waters were saltier (about 0.05) during fall 2015 and 2016 than in the 2014 glider mission.





| Glider mission | AW | LIW | WMDW |
|---|---|---|---|
| ABACUS 1.1 September 2014 | 14.44 ≤ θ ≤ 27.01 °C 36.56 ≤ S ≤ 37.98 | 13.21≤ θ ≤ 13.35 °C 38.48 ≤ S ≤ 38.52 | 12.91 ≤ θ ≤ 13.17 °C 38.44 ≤ S ≤ 38.49 |
| ABACUS 1.2 Nov - Dec 2014 | 15.82 ≤ θ ≤ 18.99 °C 36.73 ≤ S ≤ 37.34 | 13.29≤ θ ≤ 13.35 °C 38.49 ≤ S ≤ 38.53 | 12.92 ≤ θ ≤ 13.17 °C 38.44 ≤ S ≤ 38.49 |
| ABACUS 2 October 2015 | 16.11 ≤ θ ≤ 23.88 °C 36.52 ≤ S ≤ 38.04 | 13.32≤ θ ≤ 13.51 °C 38.53 ≤ S ≤ 38.59 | 12.99 ≤ θ ≤ 13.17 °C 38.49≤ S ≤ 38.54 |
| ABACUS 3 Nov - Dec 2016 | 15.18≤ θ ≤ 20.64 °C 36.51 ≤ S ≤ 37.84 | 13.29≤ θ ≤13.51°C 38.49≤ S ≤ 38.63 | 13.09 ≤ θ ≤ 13.23°C 38.49 ≤ S ≤ 38.55 |


Table 2. Temperature and salinity range values measured for Atlantic Water (AW), Levantine
Intermediate Water (LIW) and West Mediterranean Deep Water (WMDW) during ABACUS 1,
ABACUS 2 and ABACUS 3 missions.

The main properties of the water masses, as well as their spatial and vertical variability and
distribution can be successfully observed through the analysis of vertical transects.
Figures 8 and 9 show the vertical sections along latitude of temperature, salinity, oxygen
concentration, CHLA concentration and turbidity collected along a transect of the ABACUS 3
mission. This North-South transect was realized from December 1st to December 9th 2016 along
the Sentinel 3 groundtrack number 713.
Temperature and salinity data collected in the surface layer (fig. 8 a,b) show a clear signature of
the AC presence. Lower temperatures and salinities are registered in the southern part of the
transect, highlighting the presence of AW recently entering the Mediterranean Sea. On the other
hand, the northern part of the transect is characterized by more saline and warm waters with
typical Mediterranean properties. Oxygen concentration in the surface layer presents the
expected distribution, without any significant latitude pattern (Fig. 8c)
The North-South pattern is again evident in the CHLA concentration section (fig. 8d). An increase
in the chlorophyll signal is registered in correspondence of the AC system at 50 m depth and, with
lower intensity, at the northern edge of the transect where terrestrial nutrient input from the
Mallorca island can be more important.
The CHLA concentration increase in the southern part of the transect shows the presence of a
lower concentration area at about 37.3°N. This signal may be associated to meanders or filaments
of the AC that deviate from the Eastward pattern of the current and impact on the biological
properties of the water masses.
The same signal can also be identified in the temperature and salinity sections, even if with
reduced intensity. The deeper layers (from 200 m to 1000 m depth) are mainly characterized by
the presence of the LIW. The relatively low oxygen concentration values and increased salinity
signals registered between 300 and 500 m depth along most of the transect are a clear signature
of the LIW presence (Fig. 9).

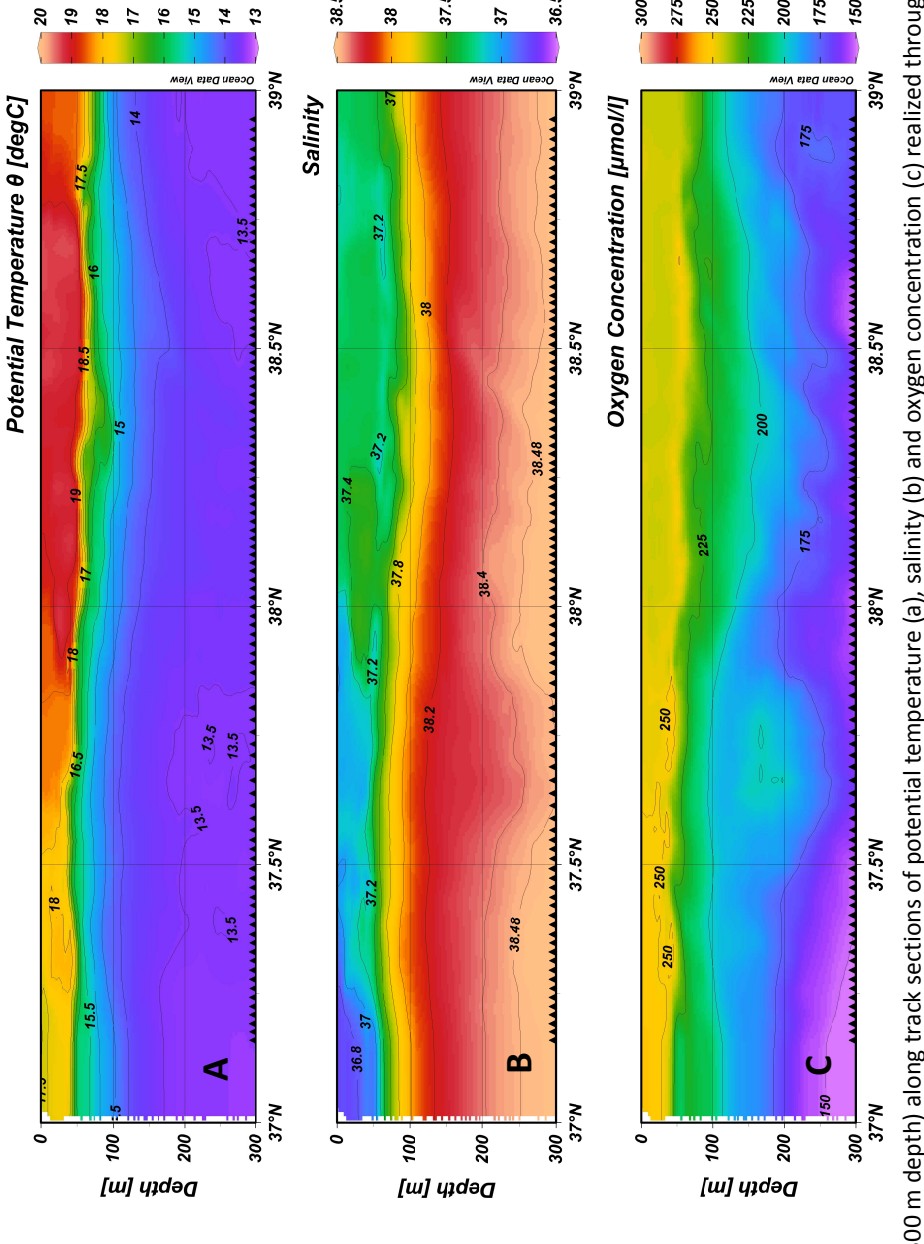

Figure 8. Surface layer (0-300 m depth) along track sections of potential temperature (a), salinity (b) and oxygen concentration (c) realized through ABACUS 3 glider data. Black triangles indicate the position of the single glider profiles.




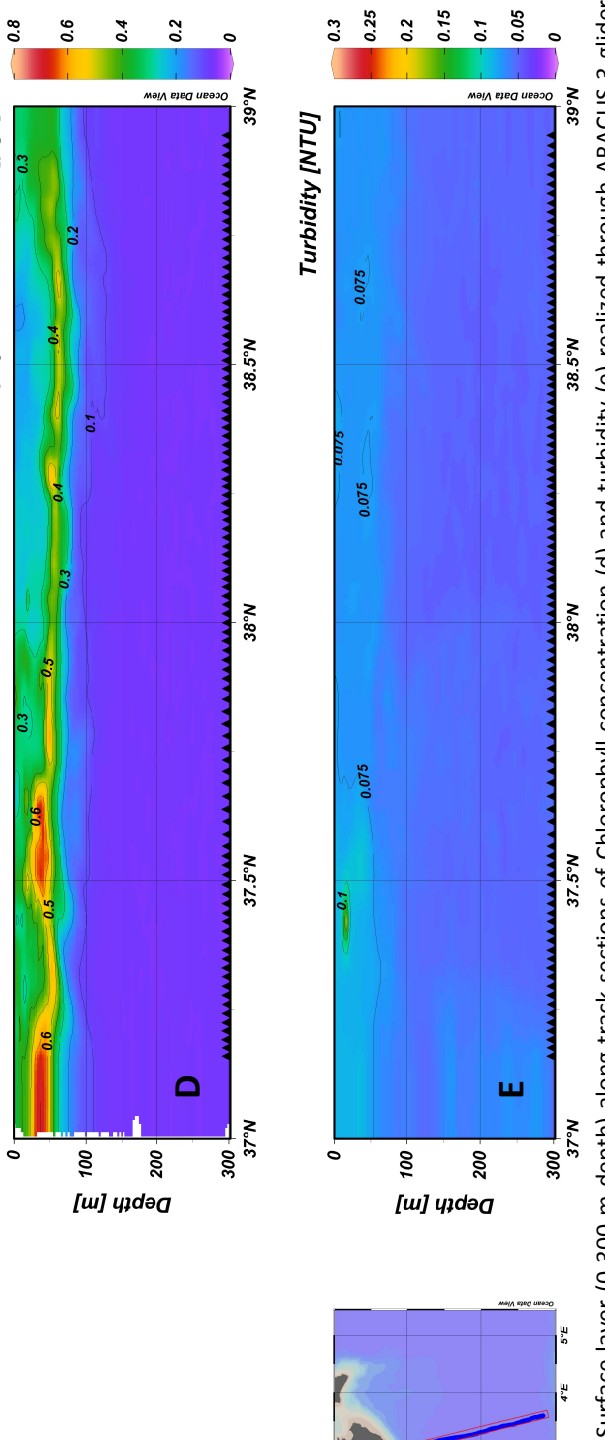

Figure 8. Surface layer (0-300 m depth) along track sections of Chlorophyll concentration (d) and turbidity (e) realized through ABACUS 3 glider data. Black triangles indicate the position of the single glider profiles.


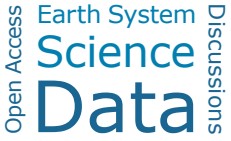

Figure 9. Intermediate and deep layer (200-1000 m depth) along track sections of potential temperature (a), salinity (b) and oxygen concentration (c) realized through ABACUS 3 glider data. Black triangles indicate the position of the single glider profiles.



The 800-1000 m depth layer, also shows the presence of a small scale structure at about 37.3°N.
In this layer, a lower temperature and salinity signal associated to increased oxygen concentration
can be found from 600 m to 1000 m depth. This structure may be associated to the signals
observed in the surface layer and surely deserves further investigations. All these data confirm the
glider ability to describe the main water mass properties at all scales from basin to mesoscale and
capture small scale structure along the water column and the investigated track. An example of
mesoscale variability analyzed through ABACUS glider data can be found in Cotroneo et al. (2016)
which present the vertical sections of multiparametric observations collected across a mesoscale
eddy.

## 5    Conclusions

The Mediterranean Sea is known to be particularly sensitive to changes in external forcings, thus
being one of the most responsive areas to climate change (Schroeder et al., 2017; Gualdi et al.,
2013) and its waters have already shown the presence of significant trends even in the deepest
layers (Fusco et al., 2008; Budillon et al., 2009). In this framework, the ABACUS dataset provides
about 1700 complete casts along the water column down to 975 m depth across the Algerian
Basin one of the key areas of the western Mediterranean where monitoring programmes should
be improved and intensified.
In particular, here we present data collected during a series of glider missions carried out from
2014 to 2016 in the framework of the ABACUS project. This effort allowed the collection of a large
dataset of physical (temperature and salinity) and biochemical (oxygen concentration, turbidity
and chlorophyll concentration) high resolution in situ observations. The reliability of these
measurements was tested and assessed through different quality control procedures, as well as
through comparisons with available historical datasets.
This dataset is available through an unrestricted repository at https://doi.org/10.25704/b200-3vf5,
where NetCDF files including different elaboration levels (from 0 to 2) and documentation are
easily accessible. This multiparametric dataset is expected to be particularly beneficial to
oceanographic studies focusing on the characterization of the hydrographic and biochemical
structure of the Western Mediterranean at all spatial scales. In fact, the presence of AW (at
different modification stages), LIW and WMDW, as well as their interannual variability, can be
observed and analysed, as reported by Aulicino et al. (2018). They included part of the presented
dataset in their multiplatform analyses, stressing the usefulness of glider repeated monitoring in
combination with altimetry and numerical simulations. Still, these observations already proved
their contribution to the analyses of the mesoscale and sub-mesoscale processes in the study
region (Cotroneo et al., 2016) whose study needs to be based on an appropriate high resolution in
situ dataset to possibly be coupled with satellite remote sensing data.
Then, we believe that the ABACUS glider dataset represents a valid unrestricted product which
could partially fill the lack of information in the AB, and a valuable tool for improving, together
with similar information collected in the framework of other AUV projects, our knowledge about
the dynamics of the Western Mediterranean and its physical and bio-chemical characteristics at
different spatial scales.
Moreover, we expect that this data could be used to improve and validate information derived
through numerical simulations, as well as to calibrate and validate present and future satellite
observations, especially those acquired through radar altimetry.
In a future perspective, cost efficient repeated glider cruises can then contribute to create a
network of endurance lines, monitoring both the short and long-term variability of the main
physical and biochemical parameters of the Mediterranean Sea. At a larger scale, the use of gliders





is crucial in the Global Ocean Observing System to fill gaps in transition regions between the open
ocean and shelf areas (Liblik et al., 2016) and to increase the array of observations in areas
traditionally lacking of in situ measurements, such as the Southern Mediterranean Sea. The
synergy between multiple gliders surveys and other data sources, i.e., satellites, models, coastal
radars, buoys and drifters, possibly including advisable occasional classical oceanographic cruises,
could represent the best strategy to be implemented.
As for the AB, the ABACUS monitoring activities are expected to be repeated and enhanced during
the coming years, taking into account the lesson learned through this study. From 2017 We
enlarged the ABACUS network and realized more glider missions per year in order to monitor both
seasonal and interannual variability along the Mallorca-Algerian coast transect.
**Data availability**
ABACUS glider data from 2014 to 2016 are available to the public in NetCDF format through an
unrestricted repository at https://doi.org/10.25704/b200-3vf5. A set of NetCDF files for data from
level 0 to level 2 is created for each glider deployment.
The SOCIB Data Centre hosts the data repository and offers a useful interface for data visualization
and download.
**ACKNOWLEDGEMENTS**

The ABACUS missions (2014) were supported by Joint European Research Infrastructure network
for Coastal Observatories (JERICO) TransNational Access (TNA) third call (grant agreement No
508  262584).
The research leading to ABACUS 3 (2016) was supported by the European Union's H2020
Framework Programme (h2020-INFRAIA-2014-2015) (grant agreement No. 654410).
Additional EU funding (PERSEUS Grant agreement no: 287600) is acknowledged.
The activities described in this paper were developed in the framework of the Italian Flagship
Project RITMARE
A.P. wishes to acknowledge support from the PRE-SWOT project (CTM2016-78607-P) funded by
the Spanish National Research Program.
SOCIB Data Server hosts ABACUS data which are available at https://doi.org/10.25704/b200-3vf5.
The authors are particularly grateful to the SOCIB Glider Facility team and the Data Centre and
Engineering and Technology Deployment staffs for their efficient cooperation.



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
