# Peer review of "Glider data collected during the Algerian Basin Circulation Unmanned Survey"

_Earth System Science Data, 2018_

## Referee Comment (RC1) · Anonymous Referee #1 · 25 Nov 2018

In this manuscript, the authors present glider datasets collected during fall 2014, 2015 and 2016. The trajectories passed along altimetry tracks (SARAL and Sentinel-3) which is very useful to perform altimetry data validation and/or to study structures with a multi-platform strategy. The campaigns took place in the Algerian Basin, which is quite under sampled and are thus an interesting contribution to study this region of the Mediterranean Sea. A complete description of the instruments as well as the quality control and data validation steps is given. Then a comparison with historical data is made and finally the authors focused on the representation of water masses characteristics by the glider. The paper is easy to read, well-structured and provide detailed description of the methods and data which are of full interest for the community. Datasets are easy to access via the given identifier and seem of high quality. I consider that this

paper must be published after minor revision. Here are some comments or questions and a few technical remarks about typographical and grammatical errors that need to be corrected.

Specific comments :

p3, lines 111-113: here you mention a change in the sampling strategy in 2014 and 2016 but in p4 line 142 it is in 2014 and 2015. Regarding Figure 2, I think that the introduction must be changed.

p4 line 133: in the introduction you write that gliders collect data in the first 975 m but here you mention that they can reach a depth of 1000 m. Why this difference?

p5 line 151: I don't understand how many glider you use. Is this a single glider which has been reused for the different missions or three distinct gliders from the same constructor and with the same characteristics ?

p5 line 162: I understood from p2 line 92 that gliders have a horizontal velocity of 0.25 m/s.

p5 line 161-162: do you have a reference or a technical sheet for the velocities?

p7 lines 241-259: I think you can add a table with the dates of the campaigns because the reader don't know exactly when they start and end. Moreover are there two campaigns during fall 2014 ? Because table 2 is split in two cases for this year.

p8 line 290: how did you filter the data?

p8 line 291: how do you average data vertically? Did you assume that one dive represent one vertical profile and one corresponding latitude or longitude ?

p8 lines 295-298: I understood from p5, line 163 that there was no data acquisition between the surface and the 20m-depth layer.

Figure 4: I understood from the text that in 2014, the glider don't reach the surface.

[Figure]

However, on the figure, you seem to have measurement from the surface down to 20 m.

Figure 5: maybe you can add as a legend the start and end dates of the profiles.

p10 line 331: I see a larger maximum mean standard deviation value on Figure 5.

p12 line 351: could you rapidly mention these standard quality control procedures?

Figure 6: Why don't you consider the saw-tooth green track in Figure 1?

p12 line 353: How many ABACUS and historical observations profiles did you get? Maybe you can indicate the number here as you did in the conclusions.

p14 line 405: is this transect representative of the others?

Technical corrections :

p2 line 66: no capital letter to "Km"

p2 line 87: no capital letter to "Km"

p2 line 92; p5 lines 161, 162: please be consistent through the whole text between cm/s or m/s

p3 line 107: please put a space between the point and "These"

p4 line 138: please replace "JERCIO" by "JERICO"

p4 line 149: please put a capital letter to "altiKa"

p5 lines 151: please remove capital letters from "Temperature, Salinity, ..."

p5 line 159: please put a space between the point and "ABACUS"

p7 line 222: no capital letter to "Kg/m$^3$"

p19 line 492: please remove the capital letter from "We"

---

## Referee Comment (RC2) · Anonymous Referee #2 · 29 Nov 2018

The paper presents data recorded by gliders in the Algerian Basin in an under-sampled area (between Balearic Island and Algerian waters). Eight transects have been performed in fall 2014 (3 transects), 2015(1 transect), 2016 (4 transects), following different altimetric tracks. Moreover, taking advantage from the adaptive sampling ability some mesoscales process where investigated, deviating from the planned route.

Parts of this dataset were already discussed in two scientific papers dealing with an Algerian eddy (ABACUS 2014) or with a concomitant use of altimetry(ABACUS 2014, ABACUS 2015). Nevertheless a rapid examination of the dataset demonstrate its potential interest for further studies (mainly mesoscale or even sub-mesocale processes).

The monitoring of the dynamics of this area is less achieved as the dataset cover only three consecutive years. Ones should encourage the authors to add progressively their

more recent data in the same data base, as they suggest in the conclusion. A quick overview on the SOCIB website reveals that similar experiment were conducted in May 2016, in Fall 2017 and in May 2018. If the authors agree, a less restrictive title should be proposed (an unrestricted time period ?) and the potential future extension of the database should be mentioned in the abstract.

Data processing and data quality check are the state of the art and well described. The paper is well written and organized. I tested the data repository and I appreciate the Netcdf file organisation. The on-line data visualisation is a little bit frustrating and not very informative. One would appreciate alternatively quick locks of the transects (on line or in an attached pdf file).

My general feeling is very positive and I recommend the publication of this paper and of the associated dataset, with minor modification.

Detailled comment :

ligne 52 : What is exactly MW ? - Modified Atlantic Water interacting with newly inflowing AW - Deeper water masses as LIW ?

Please choose a unified name for your different dataset. Sometime you wrote (ABACUS ABACUS 2, ABACUS 3 (figure 5) or ABACUS 1, ABACUS 2, ABACUS 3 (line 139) or ABACUS 2014 (line 315). Note that this classification disappears on the data repository.

---

## Author Comment (AC1) · 12 Dec 2018

The paper presents data recorded by gliders in the Algerian Basin in an under-sampled area (between Balearic Island and Algerian waters). Eight transects have been performed in fall 2014 (3 transects), 2015 (1 transect), 2016 (4 transects), following different altimetric tracks. Moreover, taking advantage from the adaptive sampling ability some mesoscales process where investigated, deviating from the planned route.

Parts of this dataset were already discussed in two scientific papers dealing with an Algerian eddy (ABACUS 2014) or with a concomitant use of altimetry (ABACUS 2014, ABACUS 2015). Nevertheless, a rapid examination of the dataset demonstrates its potential interest for further studies (mainly mesoscale or even sub-mesocale processes).

The monitoring of the dynamics of this area is less achieved as the dataset cover only three consecutive years. Ones should encourage the authors to add progressively their more recent data in the same data base, as they suggest in the conclusion. A quick overview on the SOCIB website reveals that similar experiments were conducted in May 2016, in Fall 2017 and in May 2018. If the authors agree, a less restrictive title should be proposed (an unrestricted time period?) and the potential future extension of the database should be mentioned in the abstract.

Data processing and data quality check are the state of the art and well described. The paper is well written and organized. I tested the data repository and I appreciate the Netcdf file organisation. The on-line data visualisation is a little bit frustrating and not very informative. One would appreciate alternatively quick locks of the transects (on line or in an attached pdf file).

My general feeling is very positive and I recommend the publication of this paper and of the associated dataset, with minor modification.

We thank the Reviewer for his work and comments, and we are glad that he appreciated this version of our manuscript.

We agree with him on the possibility of removing the reference years from the title and will contact the Editor about it.

A new sentence about data extension has been added to the abstract as more recent data (years 2017/2018) will be added to the same DOI database soon.

Suggestions from the Reviewer on the online visualization of the data have been forwarded to the SOCIB Data Centre to improve ABACUS plot accessibility.

A point by point response (in red) to the specific comments received is provided below.

Best regards, Dr Yuri Cotroneo

Detailled comment:

line 52: What is exactly MW? - Modified Atlantic Water interacting with newly inflowing AW - Deeper water masses as LIW?

At line 52, we used the acronym MW to refer to all water masses that are typically formed or resident in the Mediterranean Sea.
The acronym was removed to clarify.

Please choose a unified name for your different dataset. Sometime you wrote (ABACUS ABACUS 2, ABACUS 3 (figure 5) or ABACUS 1, ABACUS 2, ABACUS 3 (line 139) or ABACUS 2014 (line 315). Note that this classification disappears on the data repository.

The text has been corrected and the "ABACUS 1, ABACUS 2, ABACUS 3" format has been adopted for the different missions. "ABACUS" will now describe the entire research programme.

---

## Author Comment (AC2) · 12 Dec 2018

We are glad that the referee appreciated our effort in providing a useful and carefully quality checked dataset for main water column parameters in the Algerian Basin.

The manuscript will be improved accepting all the received suggestions and the revised version will be provided soon. According to suggestions received from both reviewers, some sentences in the text and even the title have been updated. A point by point response (in red) to the specific comments received is provided below.

Best regards, Dr Yuri Cotroneo

Please also note the supplement to this comment:

https://www.earth-syst-sci-data-discuss.net/essd-2018-130/essd-2018-130-AC2-supplement.pdf

[Figure]

**Supplement:**

In this manuscript, the authors present glider datasets collected during fall 2014, 2015 and 2016. The trajectories passed along altimetry tracks (SARAL and Sentinel-3) which is very useful to perform altimetry data validation and/or to study structures with a multi-platform strategy. The campaigns took place in the Algerian Basin, which is quite under sampled and are thus an interesting contribution to study this region of the Mediterranean Sea. A complete description of the instruments as well as the quality control and data validation steps is given. Then a comparison with historical data is made and finally the authors focused on the representation of water masses characteristics by the glider. The paper is easy to read, well-structured and provide detailed description of the methods and data which are of full interest for the community. Datasets are easy to access via the given identifier and seem of high quality. I consider that this paper must be published after minor revision. Here are some comments or questions and a few technical remarks about typographical and grammatical errors that need to be corrected.

We are glad that the referee appreciated our effort in providing a useful and carefully quality checked dataset for main water column parameters in the Algerian Basin.

The manuscript will be improved accepting all the received suggestions and the revised version will be provided soon.

According to suggestions received from both reviewers, some sentences in the text and even the title have been updated.

A point by point response (in red) to the specific comments received is provided below.

Best regards, Dr Yuri Cotroneo

Specific comments:

p3, lines 111-113: here you mention a change in the sampling strategy in 2014 and 2016 but in p4 line 142 it is in 2014 and 2015. Regarding Figure 2, I think that the introduction must be changed.

The Reviewer is right. Sampling track was changed in 2014 and 2015. We corrected the information mentioned in the introduction paragraph.

p4 line 133: in the introduction you write that gliders collect data in the first 975 m but here you mention that they can reach a depth of 1000 m. Why this difference?

This difference is linked to the beginning of the ascending procedures of the gliders when reaching its maximum depth and to the existence of a security margin of 25 m. In fact, a very well ballasted glider could need about 10 meters to change the direction of the vertical motion (sign change in diving-speed in meters/second). Technically, 975m is the 'diving target depth' meaning the glider will begin the bottom-inflection when reaching that depth so it won't stop descending at that depth but will need 10-15 meters to begin ascending. Accordingly, we effectively collect data down to 975 m depth, where the lower inflection depth is located, while the glider has its maximum navigation depth at about 1000 m. In the new version of the manuscript we highlight that data collection is always limited to the maximum depth of 975 m (i.e. lines:41, 111, 132, 151, 194, 332, 414, 447, 460, 462, 464).

p5 line 151: I don't understand how many gliders you use. Is this a single glider which has been reused for the different missions or three distinct gliders from the same constructor and with the same characteristics?

We thank the Reviewer for this comment. The 2014 and 2016 missions were realized with the same glider (named SLDEEP 001), while the 2015 mission was realized with a different glider (named SLDEEP000) from the same constructor, with the same characteristics and undergoing the same calibration procedures.
The name of the glider used for each cruise can be found in the dataset page, as well as in the file name of the data available for each mission.
A new sentence has been added in the text at line 248:
"All these operations were performed on the two SLOCUM gliders used during the ABACUS missions (SLDEEP001 for 2014 and 2016; SLDEEP000 for 2015) as part of the pre-missions tests, and contributed in assessing the quality of the collected data."

p5 line 162: I understood from p2 line 92 that gliders have a horizontal velocity of 0.25 m/s.

Horizontal speed usually ranges between 0.35 and 0.50 m/sec but this is very variable and context dependent.
Velocity data reported at line 94 are referred to a general range of glider velocities, while values reported at line 161 are calculated by trigonometry considering vertical speed and diving angle for the specific glider sampling configuration adopted during ABACUS missions. Text has been modified at line 94 to describe the wider range of horizontal velocities that the glider can reach excluding any influence from ocean currents.
A reference has now been added, also considering the following comment by the Reviewer.

p5 line 161-162: do you have a reference or a technical sheet for the velocities?

Some details about glider velocities are available in:
- Griffiths, G., Ed., Davis, R. E., C. C. Eriksen, and C. P. Jones, 2002. Autonomous buoyancy-driven underwater gliders, In: Technology and Applications of Autonomous Underwater Vehicles, Taylor and Francis, London.
- Rudnick, DL, Davis RE, Eriksen CC, Fratantoni DM, Perry MJ. 2004. Undersea gliders for ocean research. Marine Technology Society Journal. 38:73-84.
- Jones, C., E. Creed, S. Glenn, J. Kerfoot, J. Kohut, C. Mudgal, and O. Schofield, 2005. Slocum gliders — A component of operational oceanography. Proc. 14th Int. Symp. on Unmanned Untethered Submersible Technology, Lee, NH, Autonomous Undersea Systems
- Merckelbach, L., D. Smeed, and G. Griffiths, 2010. Vertical Water Velocities from Underwater Gliders. J. Atmos. Oceanic Technol., 27, 547–563, https://doi.org/10.1175/2009JTECHO710.1

p7 lines 241-259: I think you can add a table with the dates of the campaigns because the readers don't know exactly when they start and end. Moreover, are there two campaigns during fall 2014? Because table 2 is split in two cases for this year.

The Reviewer is right. During 2014 two separate legs were realized and the glider was retrieved on land between the two legs. In the new version of the manuscript, we added the specific dates of the missions in the text (lines 254-278). Additionally, we added date information in Table 2, were the indication of the months for each mission was already shown.

p8 line 290: how did you filter the data?

Filters applied during the generation of level 1 and Level 2 data by the SOCIB DATA CENTRE are described in a series of internal report and published documents (e.g. Troupin et al., 2016).
In level 1 and 2, filters are mainly focused on the processing of pressure data and on the interpolation of missing values. Pressure is filtered using a low pass filter described in the Seabird Data Processing Manual.
- Troupin, C., Beltran, J.P., Heslop, E., Torner, M., Garau, B., Allen, J., Ruiz, S., and Tintoré, J.: A toolbox for glider data processing and management. Meth. Oceanogr. 13–14. http://dx.doi.org/10.1016/j.mio.2016.01.001, 2016.

p8 line 291: how do you average data vertically?

Did you assume that one dive represents one vertical profile and one corresponding latitude or longitude?

According to Troupin et al. (2016), whose toolbox is at the basis of the SOCIB Data Centre dataset production, Level 1 contains processed glider data, as sequences of measurements along the glider trajectory, with interpolated position coordinates to match the times of measurement by the sensor, and with unit conversions and filters applied. Level 2: contains gridded glider data, which means that the glider data are interpolated onto a user configured grid in the vertical (1m in the ABACUS project dataset) and stored as vertical profiles. The profiles are obtained by interpolation of level 1 data to produce regular homogeneous and

instantaneous profiles from each up or downcast, using the mean time and position of the corresponding cast for the profile location and time.

p8 lines 295-298: I understood from p5, line 163 that there was no data acquisition between the surface and the 20m-depth layer.

We thank the Reviewer for this comment, as it gave us the opportunity to better explain the data sampling strategy of ABACUS. The data collection in the first 20 m of the water column was realized during each glider mission, but with a strategy that changed during the three years of the project according to the need of improving the sampling in this layer.
In particular, in 2014 we collected data in the 0-20 m layer every 4th profile with a final spatial resolution of about 8.4 km. This resolution, relatively low for glider operations, is the result of the combined effect of the data collection (limited at downcast only) and the navigation strategy (breaking the surface every second couple of ascending/descending profiles) adopted in 2014.
In 2015, we collected a complete cast from surface to bottom about every 5 km. This was obtained acquiring data on both downcast and upcast.
Finally, in 2016 we tested the glider possibility to sample the entire water column from 0 to 1000 m depth at the resolution of 2.8 km. This was obtained sampling during both upcast and downcast and setting the glider to break the surface after every ascending cast. The experiment was successful, but limited in time due to battery constraints. For this reason, during ABACUS 3, the layer between 0 and 20 m depth is sampled at a variable resolution between 2.8 and 5 km.
The text has now been improved with a better description of the data collection in the very surface layer (lines 161-175). Figure 3 has been modified accordingly.

Figure 4: I understood from the text that in 2014, the glider did not reach the surface. However, on the figure, you seem to have measurement from the surface down to 20m.

Figure 4 includes data in the first 20 m depth as data acquisition in this layer was realized each year, but was not sufficiently described in the text.
Now the sampling in the layer 0-20 m is better explained in the text at lines 161-175 also according to the previous comment from the Reviewer.

Figure 5: maybe you can add as a legend the start and end dates of the profiles.

Starting and ending date of each transect, resulting in mean profiles showed, are now reported in figure 5.

p10 line 331: I see a larger maximum mean standard deviation value on Figure 5.

The reviewer comment is probably generated by a lack of explanation about the numbers reported in the text and of the standard deviation values plotted in figure 5.
The standard deviation reported in the text at lines 344-350 have been calculated through a mean of the standard deviations calculated at every single depth among all the available profile for each transect. These means involve all depths, generating low values of mean standard deviation.

On the other hand, shadowed values in figure 5 are the standard deviation values calculated at each depth among all the available profile for each transect.

Aiming at comparing the magnitude of the natural variability and the instrument precision, we consider that comparing values of line 344-350 is the worst hypothesis, (very low mean natural standard deviation), thus enforcing the idea that glider in situ measurements are precise enough to represent the basin variability.

In the new version of the manuscript, a deeper description of the standard deviation values showed in figure 5 is now included to better explain the difference between what is reported in text and figure. Standard deviation values reported in figure 5 are also commented at lines 350-352.

p12 line 351: could you rapidly mention these standard quality control procedures?

The text has been modified mentioning the spike removal and the comparison between adjacent profiles as standard QC procedures.

Figure 6: Why don't you consider the saw-tooth green track in Figure 1?

The Reviewer is right. Data were missing.
Data have been added in figure 6 and 7

p12 line 353: How many ABACUS and historical observations profiles did you get?
Maybe you can indicate the number here as you did in the conclusions.

The number of ABACUS and historical profiles have been added and updated according to the addition of the data from the saw-tooth glider sampling in figures 6 and 7.
The manuscript has been modified to include the updated number of ABACUS and historical profiles both in this paragraph and in the conclusions.

p14 line 405: is this transect representative of the others?

Yes, it is representative of the general data distribution and ocean circulation of the study area. Additionally, it is the most recent transect available in the dataset. This is the main reason why it was chosen.
This information has been added to the text.

Technical corrections:

p2 line 66: no capital letter to "Km"
Corrected

p2 line 87: no capital letter to "Km"
Corrected

p2 line 92; p5 lines 161, 162: please be consistent through the whole text between cm/s or m/s
Corrected

p3 line 107: please put a space between the point and "These"
Corrected

p4 line 138: please replace "JERCIO" by "JERICO"
Corrected

p4 line 149: please put a capital letter to "altiKa"
Corrected

p5 lines 151: please remove capital letters from "Temperature, Salinity"
Removed

p5 line 159: please put a space between the point and "ABACUS"
Corrected

p7 line 222: no capital letter to "Kg/m3"
Corrected

p19 line 492: please remove the capital letter from "We"
Corrected